# A Dynamic Approach to Low-Cost Design, Development, and Computational Simulation of a 12DoF Quadruped Robot

Md. Hasibur Rahman [1,2], Saadia Binte Alam [1,2,*], Trisha Das Mou [1,2], Mohammad Faisal Uddin [1,2] and Mahady Hasan [1,2]

1    Department of Computer Science Engineering, IUB, Dhaka 1229, Bangladesh
2    RIoT Research Center, IUB, Dhaka 1229, Bangladesh
\*    Correspondence: saadiabinte@iub.edu.bd

**Abstract:** Robots equipped with legs have significant potential for real-world applications. Many industries, including those concerned with instruction, aid, security, and surveillance, have shown interest in legged robots. However, these robots are typically incredibly complicated and expensive to purchase. Iron Dog Mini is a low-cost, easily replicated, and modular quadruped robot built for training, security, and surveillance. To keep the price low and its upkeep simple, we designed our quadruped robot in a modular manner. We provide a comparative study of robotic manufacturing cost between our proposed robot and previously established robots. We were able to create a compact femur and tibia structure with sufficient load-bearing capacity. To improve stability and motion efficiency, we considered the novel Watt six-bar linkage mechanism. Using the SolidWorks modeling software, we analyzed the structural integrity of the robot's components, considering their respective material properties. Furthermore, our research involved developing URDF data for our quadruped robot based on its CAD model. Its gait trajectory is planned using a 14-point Bezier curve. We demonstrate the operation of the simulation model and briefly discuss the robot's kinematics. Computational methods are emphasized in this research, coupled with the simulation of kinematic and dynamic performances and analytical/numerical modeling.

**Keywords:** kinematics; simulations; quadruped robot; watt six-bar linkage; PyBullet; URDF; 12DoF

## 1. Introduction

Numerous quadrupedal robots have been developed in recent decades. The potential of mobile robots to replace people in high-risk situations has made this a fascinating area of robotics study [1], including first response, flammable and toxic substance disposal, and lidar field navigation. Most remotely operated robots fall into three categories: Those with wheels, those with a crawler-type under-carriage, and those with legs [2]. Although mobile robots that move on wheels or crawlers may be effective on the ground level, their mechanisms are severely limited by obstacles such as hills and mountains. Legged robots have a wider variety of potential applications, as they can traverse difficult terrain [3]. The number of feet may be used to categorize robots into three groups: Bipeds, quadrupeds, and multi-legged. More people are interested in quadruped robots than in the past, as they are more stable, can carry more weight than biped robots, have better mobility performance than multi-legged robots, and are more efficient at moving than the aforementioned robots [4]. Rapid and accurate evaluation of dynamic characteristics is essential for precise modeling, estimation, and control of robots. Scientists in the field of robotics can greatly benefit from having access to techniques that can reduce error and speed up the development process. In this study, we provide a computationally based, organized simulation model run in the Pybullet physics engine, in order to address a wide variety of issues that arise when constructing quadruped robots. Our research here describes the connection between a kinematics equation and the characteristics of the joints of a four-legged robot in depth. The

robot's design features legs with three degrees of freedom. in this study, we emphasize the use of computational methods, such as modeling and simulation of kinematic and practical implementations.

Bipedal and quadrupedal robots are incredibly well-known for their complex, sophisticated mechanics and mathematical techniques, although robotics researchers deal with a wide variety of robots at present. Parts of the quadruped robot are relevant to creatures with four legs. Simulating a high-quality, mechanically constructed, and hydraulically operated quadruped robot requires the employment of at least twelve degrees of freedom [5]. Scientists have previously developed robot engine controls for a quadruped robot [6]. Implementing a four-legged robot relies heavily on simulations run in software. This research suggested a simulation model with outlined components and parameters, such as kinematics, designs, and a standard robot description format. To simulate the motion of a robot's 2DoF leg, kinematics solutions have been implemented in a purpose-built 3D program [7]. One research highlight was demonstrating a novel robot design using electrically operated motors [8]. Additionally, most quadruped robots, such as wildcat [9], use a two-joint construction for their primary leg component. This framework is uncomplicated, understandable, and straightforward to manage. However, there are significant biological advantages to the three-joint limb structure of toed animals, such as cats, dogs, and lions, regarding walking velocity. MIT Cheetah utilized a three-segment construction that allowed for a running velocity of 6 m/s and the efficient passage of obstacles [10]. Additionally, the Cheetah-cub was created with a pantograph leg arrangement, in order to simplify controlling three links with only two joints and allowing it to achieve a trot [11]. Additionally, Pneupard—a genuine "cat-sized" robot—has utilized the same pantograph technique [12]. It has been shown that placing motors at the shoulder works well for high-velocity locomotion. Ming Lu has highlighted a 2DoF-based parallel leg formation [13]. A hybrid-legged wheeled robot, which uses a wheel and leg for walking, has been described in [14]. Kinematic analysis can be used as the determining element for quadruped robots, and recent studies [15,16] have detailed exquisite mathematical kinematics techniques. A motion observer study for a four-legged robot capable of traversing rugged terrain has been carried out [17]. Gait pattern creation is critical for quadruped robots, as they require precise trajectory adjustment. Sooyeong Yi has described the two-phase discontinuous gaits of quadruped walking robots [18]. Thanhtam Ho has created a biomimetic self-contained quadruped bounding robot [19].

The researcher gave an overview of a quadruped robot model developed in a dynamic simulator with an alternate gait creation mechanism [20]. Through structural simulation, one of the essential load test analyses of a robot leg has been detailed [21]. Knowing how much weight it can withstand and how long it can withstand pressure is beneficial. The researcher demonstrated a variable-based design for a quadruped robot using the equivalent motion, a validation approach, and a non-programmable method [22–24]. Previously, researchers have employed computer modeling and analysis before constructing a massive robot, thus reducing instrument losses and making the model construction process faster. MIT researchers have used 3D design control to demonstrate improved design and advancement [25].

In SolidWorks, we created a working prototype of a four-legged robot with precise joint measurements, specifications, and dimensions. Initially, we intended to construct our prototype using standard methods, where the servomotors are usually in joint areas like the coxa–femur joint and femur–tibia joint; however, the size and weight of our servomotors became assembly limiting considerations. Due to the poor weight distribution in the body and joints of our quadruped robot, the design was precarious. To address this problem, we built a Watt six-bar linkage mechanism with all servomotors in the coxa joint region and linkages connecting the femur and tibia to their respective servomotors. This structure allowed us to improve the weight distribution. Figure 1 shows the quadruped robot iron dog mini. Table 1 shows some feature comparisons between the existing and proposed robots.

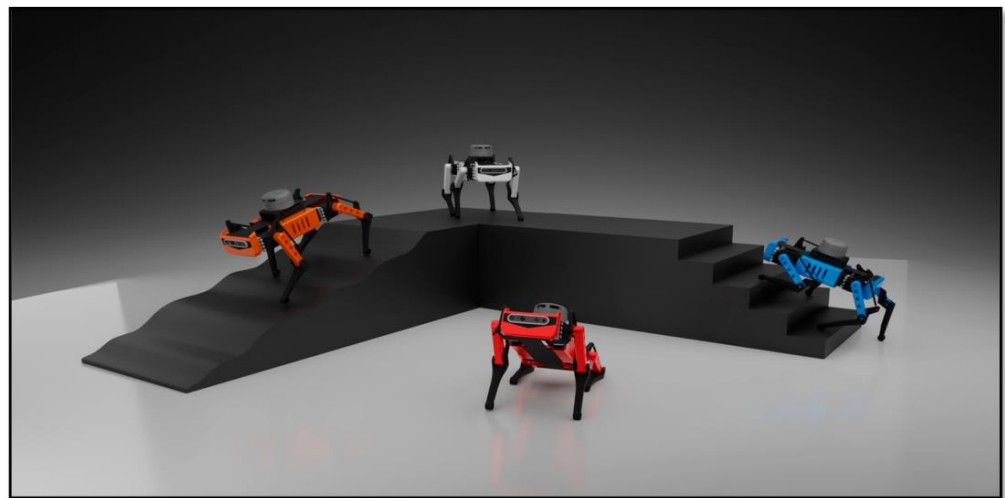

**Figure 1.** Iron Dog mini, a low-cost quadruped robot with Watt six-bar linkage mechanism.

**Table 1.** Comparison of existing models with the proposed model. The table includes the technology adoption, features, and mechanisms of previous and current research based on quadruped robots.

| Robot | Year | State-of-the-Art Technology Adoption | | | | | | | |
|---|---|---|---|---|---|---|---|---|---|
| | | Light Weight | Heavy Weight | Carry Load | Low Cost | Modular | Power Efficient | Mechanism | Agile |
| Sony Aibo | 1999 | ✓ | - | - | - | - | ✓ | Electric | - |
| BigDog [26] | 2005 | - | ✓ | ✓ | - | - | - | Hydraulic | ✓ |
| Scalf1 [6] | 2011 | - | ✓ | ✓ | - | - | - | Hydraulic | - |
| Frog [27] | 2013 | - | ✓ | ✓ | - | - | - | - | - |
| Alpha Dog [28] | 2012 | - | ✓ | ✓ | - | - | - | Hydraulic | ✓ |
| HyQ [29] | 2011 | - | ✓ | ✓ | - | - | - | Hydraulic and Electric | ✓ |
| Baby Elephant [30] | 2013 | - | ✓ | ✓ | - | - | - | Serial–parallel Hybrid | ✓ |
| AnyMal [31] | 2016 | - | ✓ | ✓ | - | - | ✓ | Electric | ✓ |
| Spot [32] | 2017 | - | ✓ | ✓ | - | ✓ | ✓ | Electric | ✓ |
| MIT Cheetah 3 [25] | 2018 | - | ✓ | ✓ | - | - | ✓ | Electric | ✓ |
| Unitree Laikago | 2017 | - | ✓ | ✓ | - | ✓ | ✓ | Electric | ✓ |
| Stoch 2 [33] | 2019 | - | ✓ | - | - | - | - | Electric-five bar linkage | ✓ |
| Proposed | 2022 | ✓ | - | ✓ | ✓ | ✓ | ✓ | Electric-Six bar Linkage | ✓ |

The contributions and motivation of this work are described below:

- We introduce the novel Watt six-bar linkage mechanism for better walking motion.
- The motivation of this research was to develop a low-cost and modular quadruped assistive robot platform for use in security and surveillance operations.
- We innovatively designed robot parts to make it modular, such that users can quickly assemble and disassemble the robot.
- We explain the kinematic equations, demonstrate the URDF process, and test several commands in the PyBullet physics engine.
- We discuss the material characteristics and structural analysis of the robot's parts.

## 2. Design Principle

Most quadruped robots have 8–16 degrees of freedom (DoF). While these are the most common, there are various other viable architectural options. The axis of all eight joints (four hips, four knees) is parallel to itself, making up all eight degrees of freedom. Although an 8-DoF robot lacks the hip joint's transverse swing flexibility, they are nonetheless easy to control and capable of fast forward and backward motion. However, the motion performance of 8-DoF quadruped robots is hindered by their poor steering capabilities and inability to carry out transverse motion. A quadruped robot with 16 DoF has more joints for agile maneuvering, but is more complicated and, hence, more difficult to control. To allow for three rotatable joints in each leg, we developed a 12 DoF design. The notion of a four-legged robot was mostly inspired by domesticated feline and canine animals, which are distinguished from other domestic species by the presence of an endoskeleton, allowing for greater movement. The development of a quadruped robot with 12 degrees of freedom is a key part of our study. Our intricate quadruped robot is made up of many different elements with rotating joints. The coxa, femur, and tibia are the most vital components of the proposed quadruped robot.

A rod end, servo arm, rod end linker, and servo horn are also included in the design. Three typical bones make up each leg: The coxa (hip bone), femur (thigh bone), and tibia (shin bone). Twelve servo actuator motors are employed to translate the links and modify the joints. The different parts of robot are shown in Figure 2.

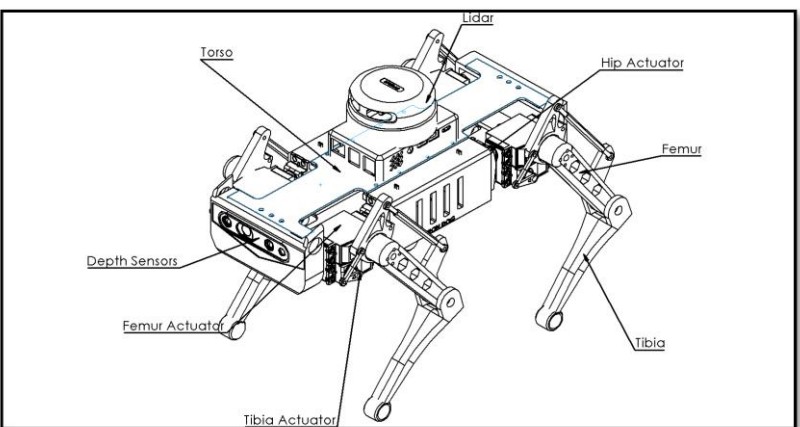

**Figure 2.** Different essential parts of the quadruped robot equipped with six-bar linkage mechanism.

The locomotion of the quadruped robot depends on the translations and rotations of the coxa joint, femur, and tibia. The kinematic analysis defines the motion of the quadruped robot's legs, where we can correlate the forward and inverse kinematics. The objective of this study is to create a quadruped robot with a body made entirely of 3D-printed parts and reasonably priced servo motors. Given its size and capabilities, the robot has been built to be able to carry a reasonable payload. As such, we designed the robot's torso in such a way that we additional components can be mounted on our robot. Our design principle allows the quadruped robot to be modular, such that the user can easily replace any broken or defective components without replacing the whole robot, massively reducing the affordability and maintenance costs of our quadruped robot. Our modular-based design is shown in Figure 3. Table 2 shows essential parameters of Iron dog mini.

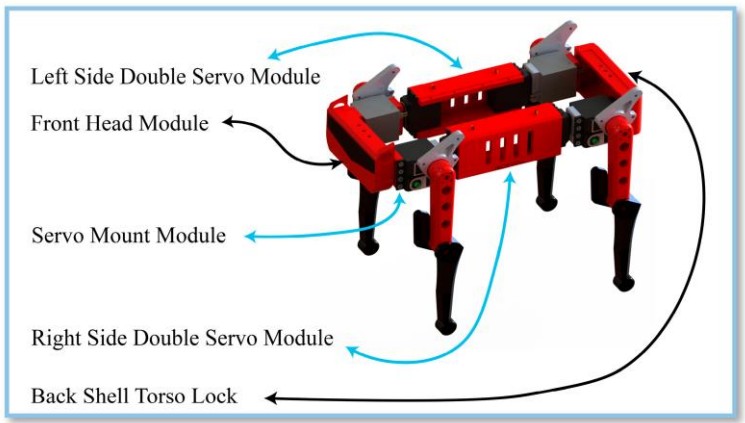

**Figure 3.** Different modules of Iron dog mini.

**Table 2.** Essential Parameters of Iron Dog mini, including the height, weight, and mass values of different links.

| Parameter | Value | | | | Total Mass (g) | Parameter | Value | | | | Total Mass (g) |
|---|---|---|---|---|---|---|---|---|---|---|---|
| | Height (cm) | Width (cm) | Number of Uses | Single Part Mass (g) | | | Height (cm) | Width (cm) | Number of Uses | Single Part Mass (g) | |
| Base | 30 | 6 | 2 | 92 | 184 | Servo Arm | 2.4 | 0.63 | 4 | 5 | 20 |
| Femur | 12 | 2.5 | 4 | 55 | 220 | Long rod end | 4.5 | 0.25 | 4 | 7 | 28 |
| Tibia | 17 | 2.5 | 4 | 48 | 192 | Short Rod end | 1.2 | 0.2 | 4 | 1.8 | 7.2 |
| Actuator | 5.6 | 2 | 12 | 80 | 960 | Front Head Module | 7 | 12 | 1 | 55 | 55 |
| Rolling Servo mount | 4 | 2 | 4 | 10 | 40 | Back Shell For torso lock | 7 | 12 | 1 | 48 | 48 |
| Side Servo Mount | 4 | 2 | 4 | 8 | 32 | Side pitch mount (double Servo Module) | 12 | 5.6 | 2 | 120 | 240 |
| Servo Horn | 0.14 | 0.025 | 4 | 6 | 24 | Screws | 0.3 | 0.05 | 74 | 1 | 74 |
| Free linker | 6.32 | 4.60 | 4 | 12 | 48 | Battery | - | - | 1 | 250 | 250 |
| Other Parts | - | - | - | 150 | 150 | Total Weight of Robot After Fully Mounted | | | | 2572.2 g | |

## 3. Working Mechanism

Iron dog mini consists of 12 servo motors that control the joint angles at the coxa, femur, and tibia. Figure 2 shows detailed information on the locations and orientations of the servo motors. Our design provides better weight balance, as we concentrated the weight close to the four corners of the robot's main body by locating the three servo motors of a single leg at the hip joint area. Actuator 1 controls the joint angle of the coxa. The hip of the robot is directly coupled with the servo motor, where the hip joint helps to stabilize the robot during motion.

Actuator 2 controls the Femur, and the femur link is directly coupled with the servo motor. Actuator 3 controls the Tibia, but the Tibia joint is not directly coupled with the servo motor. To control the Tibia link, we introduced the Watt six-bar linkage mechanism into our design, which is a unique approach. Conventional mechanisms such as four-bar linkage mechanisms have some motion constraints due to their limited design variables. We have pointed out some advantages of the six-bar linkage mechanism over the four-bar linkage mechanism.

- The watt six-bar linkage mechanism provides a greater range of motion for leg actuation than the four-bar linkage mechanism.

- The watt six-bar linkage mechanism produces leg motion during gait generation, which is very close to the leg motion of a four-legged animal compared to the four-bar linkage mechanism.
- The four-bar linkage mechanism has many motion constraints. Therefore, the four-bar linkage mechanism robot has a limited range of motion for its leg. The four-bar linkage has a total of eight design variables.
- The watt six-bar linkage mechanism has fourteen design variables.
- The six-bar linkage has more motion parameters than the four-bar linkage, increasing the range of motion.

Additionally, the gait generation produced by the watt six-bar linkage mechanism is found to be highly comparable to that of a four-legged animal, making it an ideal candidate for use in quadrupedal robots. Therefore, the six-bar linkage is a more convenient mechanism for the leg actuation of quadruped robots.

We designed and optimized the six-bar linkage mechanism to manipulate the tibia link effectively, and the bar linkages are shown in Figure 4. The Watt six-bar linkage mechanism has more stability, adequate movement, and better motion efficiency than conventional four- and five-bar linkage mechanisms. The actuator that is used has a significant impact on the performance of a quadruped robot's movement. Considering the robot's weight, the actuator must provide significant torque while maintaining a rapid response time and a compact footprint. Brushless motors are widely used, due to their excellent dynamic qualities; however, they are typically more costly and more extensive in size. Thus, we used a high-voltage servo motor which has a built-in gear reducer. Moreover, we used a metal gear servo motor, providing 35 kg torque per centimeter. The single-leg configuration is shown in Figure 5.

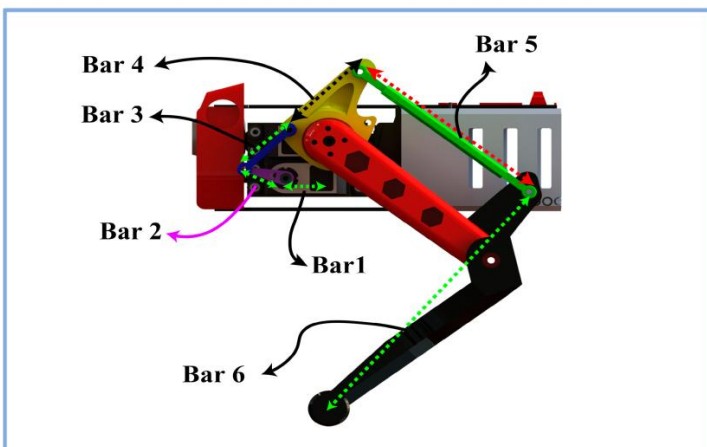

**Figure 4.** Watt Six-Bar linkage mechanism.

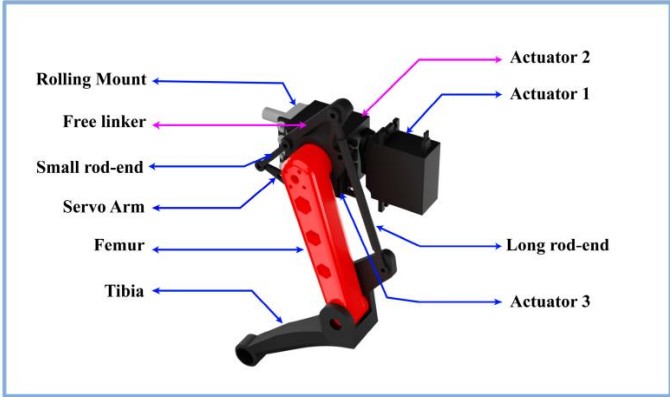

**Figure 5.** Single-leg configuration.

### 3.1. Mathematical Analysis

Quadruped robots use two types of kinematics: Forward and inverse kinematics. Robot end-effector joint positions are determined using forward kinematics, whereas manipulator joint values can be derived from inverse kinematics. In order to calculate the forward kinematics, we multiply the number of matrices. To better understand how forward kinematics operate in quadruped robots, we constructed a block-based dummy quadruped robot and figured out all the frames. In our previous work, we have demonstrated a comprehensive derivation of the true nature of forward kinematics [15]. Figure 6 depicts all the various frames and coordinate systems in detail. Additionally, Figure 6 was used to reach the foot frame, which serves as the illustration's final effector. The zeroth frame and hip joints represent the coxa coordinate $X_0$, $Y_0$, $Z_0$. To find the forward kinematics equation, we need to multiply each frame from zeroth to the fourth. If we want to reach the zeroth frame from the first frame, there will be a translation, denoted by $L_1$. When reaching the first frame from the second frame, there is no translation but, instead, a rotation, denoted by $\theta_1$. Passing from frame two to frame three involves $L_2$ and $\theta_2$ for translation and rotation, respectively. Finally, we have $L_3$ translation and $\theta_3$ rotation between the third and fourth frames. After multiple translations and rotations, we obtain the final transformation matrix, according to Denavit Hartenberg's convention. We can determine the matrix representing all possible transformations between frames 0 and 4 [15] as $T_0^4 = T_0^1 * T_1^2 * T_2^3 * T_3^4$:

$$T_0^4 = \begin{bmatrix} \phi_{11} & \phi_{12} & \phi_{13} & \phi_{14} \\ \phi_{21} & \phi_{22} & \phi_{23} & \phi_{24} \\ \phi_{31} & \phi_{32} & \phi_{33} & \phi_{34} \\ \phi_{41} & \phi_{42} & \phi_{43} & \phi_{44} \end{bmatrix}. \tag{1}$$

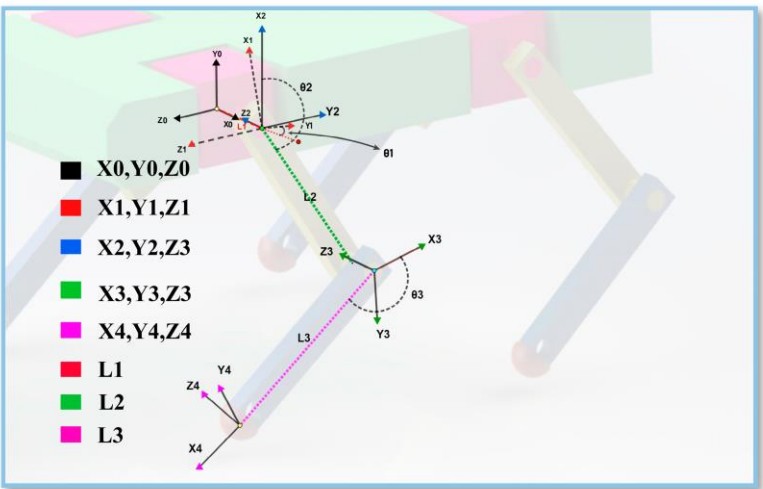

**Figure 6.** Frame-by-frame coordinate system of a quadruped robot using 3D design.

From this transformation matrix, our end effector value can be determined from the fourth column (i.e., $\phi_{14}$, $\phi_{24}$, and $\phi_{34}$). After the forward kinematics are established, inverse kinematics are required to regulate the joint settings. We sketched through the actual model for better understanding.

Based on Figure 7,

$$\theta_1 = \alpha_3 - \alpha_1. \tag{2}$$



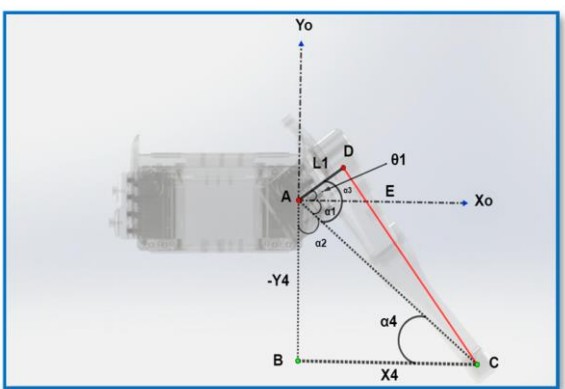

**Figure 7.** Determining $\theta_1$ utilizing the front view of the robot leg.

In $\triangle$ADC,

$$\alpha_3 = \arctan\left(\frac{\sqrt{x_4^2 + y_4^2 - L_1^2}}{L_1}\right). \tag{3}$$

In $\square$ABCE,

$$\alpha_1 + \alpha_2 = 90° \tag{4}$$

$$\Rightarrow \alpha_1 = 90° - \alpha_2. \tag{5}$$

Furthermore, in $\triangle$ABC,

$$\alpha_2 + \alpha_4 + \angle B = 180°, \tag{6}$$

$$\alpha_2 + \alpha_4 = 180° - 90° \ [\angle B = 90°], \tag{7}$$

$$\alpha_2 = 90° - \alpha_4. \tag{8}$$

Substituting this value into Equation (5) :

$$\alpha_1 = 90° - (90° - \alpha_4) \tag{9}$$

$$\Rightarrow \alpha_1 = \alpha_4. \tag{10}$$

In $\triangle$ABC,

$$\alpha_4 = arctan\left(\frac{-y_4}{x_4}\right). \tag{11}$$

Now, substituting the values of $\alpha_1$ and $\alpha_3$ into Equation (2),

$$\theta_1 = \alpha_3 - \alpha_1 \tag{12}$$

$$\Rightarrow \theta_1 = arctan\left(\frac{\sqrt{x_4^2 + y_4^2 - L_1^2}}{11}\right) - arctan\left(\frac{-y_4}{x_4}\right). \tag{13}$$

Based on Figure 8,

$$\theta_2 = -90° + \alpha_1. \tag{14}$$

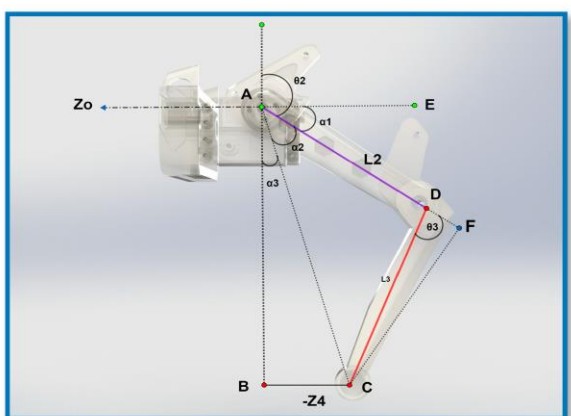

**Figure 8.** Determining $\theta_2$ utilizing the side view.

From $\square$ABFE,

$$\alpha_1 + \alpha_2 + \alpha_3 = 90° \tag{15}$$

$$\therefore \alpha_1 = 90° - \alpha_2 - \alpha_3. \tag{16}$$

From $\triangle$ABC,

$$\alpha_3 = \arctan\left(\frac{-Z_4}{\sqrt{x_4^2 + y_4^2 - L1^2}}\right). \tag{17}$$

From $\triangle ACF$,

$$\alpha_2 = \arctan\left(\frac{CF}{AF}\right). \tag{18}$$

Furthermore, from $\triangle CDF$,

$$\sin\theta_3 = \frac{CF}{CD}, \tag{19}$$

$$CF = CD\sin\theta_3, \tag{20}$$

$$CF = L_3\sin\theta_3 \, [CD = L_3], \tag{21}$$

$$\cos\theta_3 = \frac{DF}{CD}, \tag{22}$$

$$DF = L_3\cos\theta_3 \, [CD = L_3], \tag{23}$$

$$AF = AD + DF, \tag{24}$$

$$AF = L_2 + L_3\cos\theta_3 \, [AD = L_2]. \tag{25}$$

Similarly,

$$\therefore \; \alpha_2 = \arctan\left(\frac{L_3\sin\theta_3}{L_2 + L_3\cos\theta_3}\right). \tag{26}$$

Now, substituting the values of $\alpha_2$ and $\alpha_3$ into Equation (16),

$$\alpha_1 = 90° - \arctan\left(\frac{L_3\sin\theta_3}{L_2 + L_3\cos\theta_3}\right) - \arctan\left(\frac{-Z_4}{\sqrt{x_4 + y_4{}^2 - L_1^2}}\right), \tag{27}$$

$$\theta_2 = -90° + \alpha_1 \tag{28}$$

$$\Rightarrow \theta_2 = -\arctan\left(\frac{L_3\sin\theta_3}{L_2 + L_3\cos\theta_3}\right) - \arctan\left(\frac{-z_4}{\sqrt{x_4^2 + y_4^2 - L_1}}\right). \tag{29}$$

Based on Figure 9,

$$\theta_3 = 180° - \alpha. \tag{30}$$

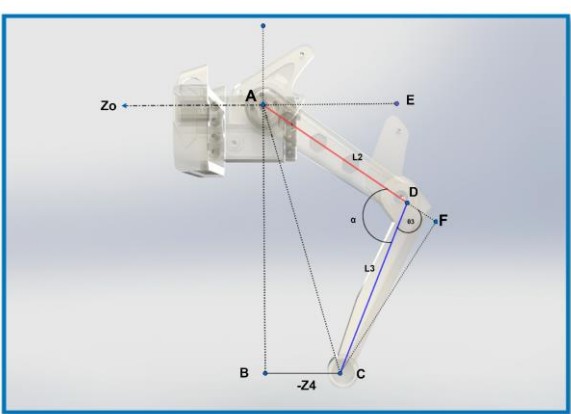

**Figure 9.** Determining $\theta_3$.

In $\triangle ACD$,

$$AC^2 = AD^2 + CD^2 - 2 \cdot AD \cdot CD \cdot \cos\alpha \tag{31}$$

$$\Rightarrow \alpha = \arccos\left(\frac{AD^2 + CD^2 - AC^2}{2 \cdot AD \cdot CD}\right). \tag{32}$$

In $\triangle ABC$,

$$AC^2 = AB^2 + BC^2, \tag{33}$$

$$AC^2 = \left(\sqrt{X_4^2 + y_4^2 + L_1^2}\right)^2 + (-z_4)^2, \tag{34}$$

$$AC^2 = x_4^2 + y_4^2 + L_1^2 + z_4^2. \tag{35}$$

Now, substituting the value of $AC^2$ into Equation (32),

$$\alpha = \arccos\left(\frac{L_2^2 + L_3^2 - X_4^2 - Y_4^2 - L_1^2 - z_4^2}{2L_2L_3}\right)\begin{bmatrix} AD = L_2 \\ CD = L_3 \end{bmatrix}, \tag{36}$$

$$\theta_3 = 180° - \arccos\left(\frac{L_2^2 + L_3^2 - x_4^2 - y_4^2 - L_1^2 - z_4}{2L_2L_3}\right). \tag{37}$$

In summary, in the derivation process of $\theta_1$, $\theta_2$ and $\theta_3$ in the inverse kinematic analysis, we took the left front leg of our quadruped robot into consideration. We positioned the robot's leg to mimic its normal standing posture (side view). Using the front view of our leg, we positioned the coxa angle, which mirrors the roll position of the robot, allowing us to identify $\theta_1$. From the two right-angle triangles ABC and ADC in Figure 7, we were able to obtain the equation for $\theta_1$ that we required. Using the profile view of our leg in Figure 8, we determined $\theta_2$. Two right-angle triangles, ABC and CDF, and one obtuse triangle, and ADC, are shown in Figure 8, and from these triangles we derived the desired equation for $\theta_2$. Using the profile of our leg, we were able to compute $\theta_3$ from Figure 9. Figure 9 depicts two triangles, one with a right-angle (ABC) and another with an acute angle (ADC), from which our desired equation for $\theta_3$ was derived.

$$\theta_1 = \arctan2\left(\left(\sqrt{x_4^2 + y_4^2 - L_1^2}\right), L_1\right) - \arctan2(-y_4, x_4), \tag{38}$$

$$\theta_2 = -\arctan2(L_3\sin(\theta_3), (L_2 + L_3\cos(\theta_3))) - arctan2\left(-z_4, \sqrt{x_4^2 + y_4^2 - L_1^2}\right), \tag{39}$$

$$\theta_3 = \pi - \arccos\left(\frac{L_2^2 + L_3^2 - x_4^2 - y_4^2 - L_1^2 - Z_4^2}{2L_2L_3}\right).\tag{40}$$

### 3.2. Simulation Model Workflow

Workflow Procedure

Next, we describe the computational method used to simulate our quadruped robots. We used the Python-based Pybullet physics engine in the Ubuntu operating system for this scientific simulation following multiple stages, as presented in Figure 10. Parts were created in the SolidWorks software using precise measurements.

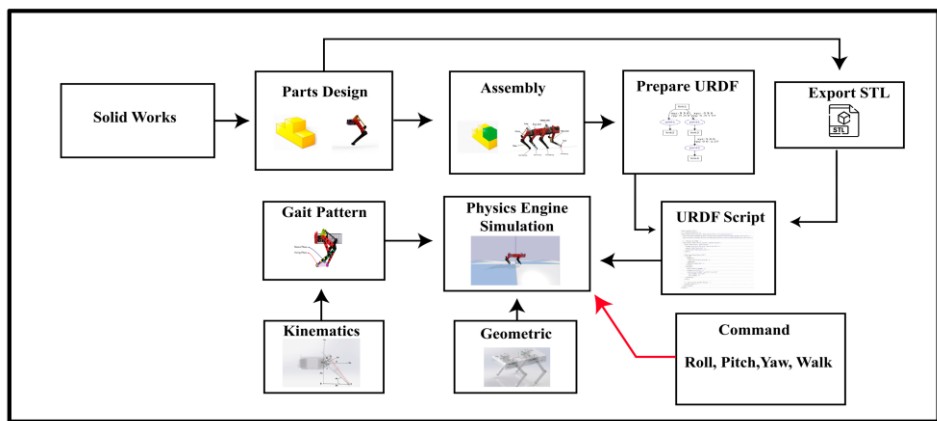

**Figure 10.** Workflow diagram for the proposed simulation model.

A great deal of thought and care must go into the design of each component before it can be assembled into a functioning robot. There are several essential parts for robots. About 64 unique parts were conceived for this study. Each of the robot's legs consists of three separate parts (the coxa, femur, and tibia), and the design is based on the principle of three degrees of freedom. Throughout the entirety of the design phase, there was a continuous process of considering the design's framework in addition to the materials that were to be utilized.

Certain tests, such as finite element modeling (FEM), finite element analysis (FoS), and strain, must be performed on the robot's core components to ascertain their durability and the maximum payload the design can withstand. In Section 4.1, we summarize relevant information in this aspect.

After making all the pieces, they must be put together in a way that allows the robot to operate well. As there are three movable and one fixed joint, each joint must be mated with extreme precision. The six-bar linkages were created to generate exterior revolute joints in addition to the leg joints. A rod-end and actuator were utilized to rotate the desired joint in the six-bar linkages.

For the simulation process, creating a URDF script was very much necessary. URDF contains all of the robot data, such as link length, link mass, material, inertia, geometry, and the path of the robot STL file. STL files are exported for every part of the robot, which contain the 3-dimensional surface geometry of the parts. This format is suitable for use with the PyBullet physics engine.

After that, we had to link our robot's gait pattern to the kinematics algorithm. The various link parameters and joint angles were contained in the Kinematics Algorithm, which provides 3D foot positions for us. We tested several gait patterns. To develop a walking motion for a quadruped robot, kinematics is insufficient. Gait generation is required to operate the robot in both real and virtual simulations. There are various gait patterns available, including the creep, trot, pace, cantor, and gallop gaits. We created a creep gait, trot gait, and pace gait for our simulation investigation, through investigating the walking patterns of genuine dogs and cats. Manually adjusting the arms and joints

of the robot to the appropriate angle is challenging, although it is plausible to adjust the motor inclination by issuing repeated instructions, this is considered inappropriate

A kinematics-based system for controlling both virtual and actual robots was created as part of this research project. Using the simulation model, complete control can be exerted over the angle of each joint, as well as the foot locations and the end effectors. This method incorporates both forward and inverse kinematics. The values of the end effectors are provided as output after all the robot's connection parameters, and the manipulator's required angle is considered.

We can also set the end effector value to achieve the appropriate location. One of the most difficult challenges is taking control of a quadruped robot from its center of gravity. If the four-legged equations are not linked to the center of gravity point, the robot will never be able to stand on its own. For example, if the robot's rear side is heavier than its front side, then the device will always be unable to walk, as it will always fall on its back. To solve this problem, we first located the center points of the robot and then directly connected them to each leg. After completing all the stages (i.e., Part design, Assembly, URDF creation, Gait pattern, and kinematics setting), we executed all the scripts for simulation using the PyBullet physics engine. We connected our implemented robot with the simulation model during the simulation process. The hardware sensors transmitted sensory data every 20 milliseconds. We can control the simulation remotely, either using a controller or giving a command.

### 3.3. Step Trajectory and Gait Generation

There are various different gait patterns used by quadruped robots, including creep, trot, and gallop, among others. Creep and trot are often employed and swapped, based on the surroundings and the desired pace of mobility. Trot serves as the most frequent gait. The two diagonal legs move in tandem, with the left front leg and right back leg moving first and the right front leg and left back leg moving second. According to the trot diagram in Figure 11, the trot is a type of gait in which two legs are activated in tandem. At one moment, the front right leg and the back left leg swing, while the right back leg and left front leg form a stance. Two motions are present: Swing and stance. The stance phase means that the toe has contact with ground; during the swinging phase, the toe is in the air.

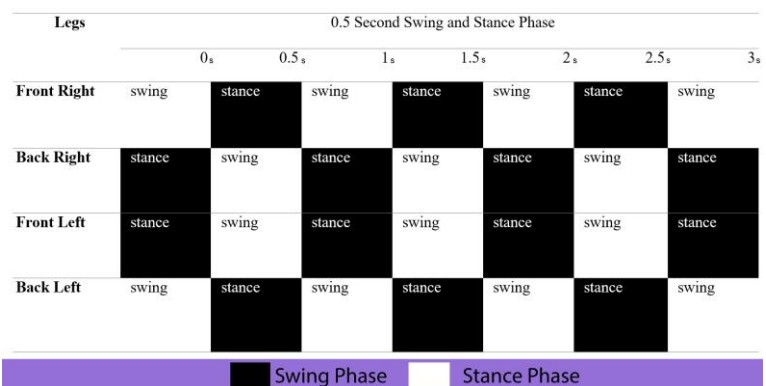

**Figure 11.** Trot Gait Patterns.

We used the Bernstein polynomial and Bezier curve for our robot's trajectory generation [34]. We created a 14-point Bezier curve to control the footstep trajectory. The control points that make up the Bezier curve were used to build the swing phase. Equation (41) shows the method for calculation of the binomial factor (d, l), and Equation (42) is used to simplify the Bezier curve calculation, where K is the current point index; t is the time; and c denotes the coordinate. Here, we have $P_0(a_0, d_0)$, $P_1(a_1, d_1)$, $P_2(a_2, d_2)$, $P_3(a_3, d_3)$, $P_4(a_4, d_4)$, $P_5(a_5, d_5)$, $P_6(a_6, d_6)$, $P_7(a_7, d_7)$, $P_8(a_8, d_8)$, $P_9(a_9, d_9)$, $P_{10}(a_{10}, d_{10})$, $P_{11}(a_{11}, d_{11})$, $P_{12}(a_{12}, d_{12})$, $P_{13}(a_{13}, d_{13})$ as the 14 control points. Equations (43) and (44) are used for curve construction. In Figure 12a, we show the trajectory steps of swing and stance, where

t = 0.5 seconds; in Figure 12b, the parametric Bezier curve is shown (the curve was plotted using MATLAB). We show all the control point values in Table 3.

$$f(d, l) = \frac{d!}{K!(d-l)!}, \tag{41}$$

$$b(t, K, c) = f(N, K) \cdot (1-t)^{(N-K) \cdot t^K \cdot c}, \tag{42}$$

$$
\begin{aligned}
a = (b(t, 0, a_0) + \quad & b(t, 1, a_1) + b(t, 2, a_2) + b(t, 3, a_3) + b(t, 4, a_4) + b(t, 5, a_5) \\
& + b(t, 6, a_6) + b(t, 7, a_7) + b(t, 8, a_8) + b(t, 9, a_9) \\
& + b(t, 10, a_{10}) + b(t, 11, a_{11}) + b(t, 12, a_{12}) + b(t, 13, a_{13})),
\end{aligned} \tag{43}
$$

$$
\begin{aligned}
d = (b(t, 0, d_0) + \quad & b(t, 1, d_1) + b(t, 2, d_2) + b(t, 3, d_3) + b(t, 4, d_4) + b(t, 5, d_5) \\
& + b(t, 6, d_6) + b(t, 7, d_7) + b(t, 8, d_8) + b(t, 9, d_9) \\
& + b(t, 10, d_{10}) + b(t, 11, d_{11}) + b(t, 12, d_{12}) + b(t, 13, d_{13})).
\end{aligned} \tag{44}
$$

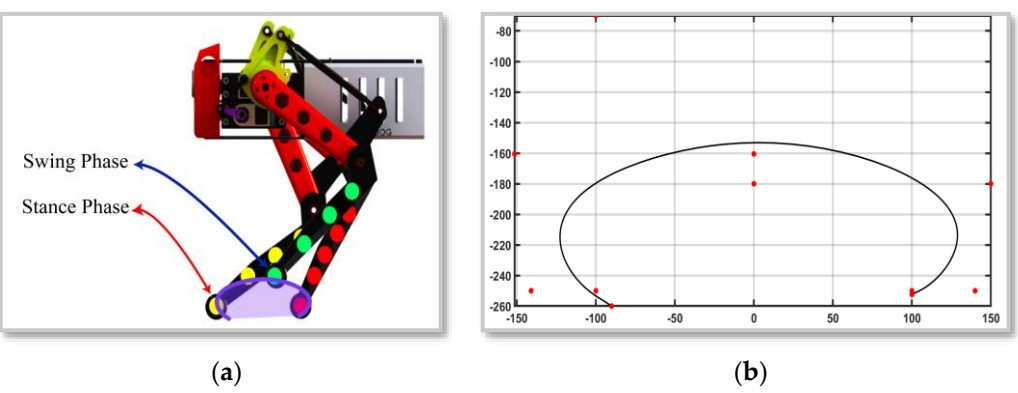

|  | (a) |  | (b) |

**Figure 12.** (**a**) Step trajectory stance and swing phase; (**b**) trajectory generation using 14-point Bezier curve.

**Table 3.** Control Points for Bezier curve.

| Control Points |  | (mm) |  | (mm) |
| --- | --- | --- | --- | --- |
| $p_0$ | $a_0$ | 100 | $d_0$ | $-252.4$ |
| $p_1$ | $a_1$ | 100 | $d_1$ | $-250$ |
| $p_2$ | $a_2$ | 140 | $d_2$ | $-250$ |
| $p_3$ | $a_3$ | 150 | $d_3$ | $-180$ |
| $p_4$ | $a_4$ | 150 | $d_4$ | $-180$ |
| $p_5$ | $a_5$ | 150 | $d_5$ | $-180$ |
| $p_6$ | $a_6$ | $-100$ | $d_6$ | $-70$ |
| $p_7$ | $a_7$ | 0 | $d_7$ | $-180$ |
| $p_8$ | $a_8$ | 0 | $d_8$ | $-160.5$ |
| $p_9$ | $a_9$ | $-151.5$ | $d_9$ | $-160.5$ |
| $p_{10}$ | $a_{10}$ | $-151.5$ | $d_{10}$ | $-160.5$ |
| $p_{11}$ | $a_{11}$ | $-141$ | $d_{11}$ | $-250$ |
| $p_{12}$ | $a_{12}$ | $-100$ | $d_{12}$ | $-250$ |
| $p_{13}$ | $a_{13}$ | $-90$ | $d_{13}$ | $-260$ |

## 4. Results and Discussion

### 4.1. Material Analysis

As our quadruped robot was designed with a low cost in mind, we had to choose a material for 3D printing that is both cheap and strong for fabrication. Therefore, we chose ABS plastic (Acrylonitrile butadiene styrene) as our printing material. While designing and developing our low-cost quadruped robot, we used SolidWorks to create around 64 distinct components. We conducted load simulation in SolidWorks for various portions (i.e., femur and tibia) of our quadruped robot, in order to better understand the strength and stiffness of our design and the material used. We obtained analytical data, such as the strain, displacement, and Factor of Safety (FoS) for the femur and tibia of our four-legged robot throughout the simulation process. By looking at these data, we could make changes to how the components were made, thus improving their performance in the long run. Table 4 shows the mechanical properties of abs plastic.

**Table 4.** Mechanical Properties of ABS Plastic.

| Property | Value | Unit |
|:---:|:---:|:---:|
| Tensile strength | 30 | MPa |
| Mass density | 1020 | $kg/m^3$ |
| Elastic modulus | 2000 | MPa |
| Shear modulus | 318.9 | MPa |
| Poisson's ratio | 0.394 | N/A |

In SolidWorks, we have utilized our femur and tibia designs to perform load analysis simulation. During load simulation, we considered approximately 7.5 kg of payload on our quadruped robot. The weight of our quadruped robot is around 2.752 kg. So, with the added payload, the total weight would be close to 10.25 kg, equivalent to a 100 N load. During the simulation process, we have obtained diagrams showing the performance of our designed parts under load. In Figure 13a,b, the displacement diagrams for the femur and tibia parts are shown, with the scale on the right indicating the degree of deformation of the components (in mm). We can see that, under a 100 N load on the femur, the maximum deformation was $6.596 \times 10^{-3}$ mm. When the 100 N load was applied on the Tibia, the maximum deformation was $5.254 \times 10^{-1}$ mm. For visual representation of the deformation, the components are highlighted with multiple colors, indicating the most deformed area (red color) and the least deformed area (blue color), according to the scale. Note that the deformation scale was adjusted for clearer visual analysis.

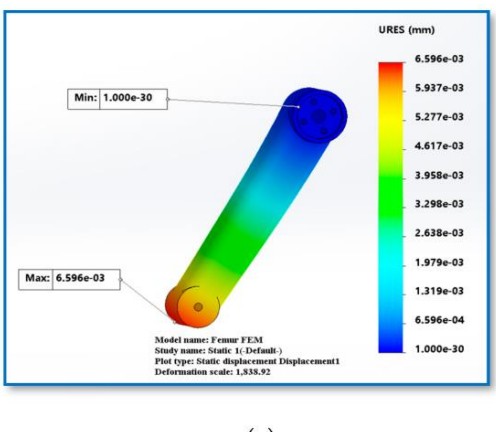
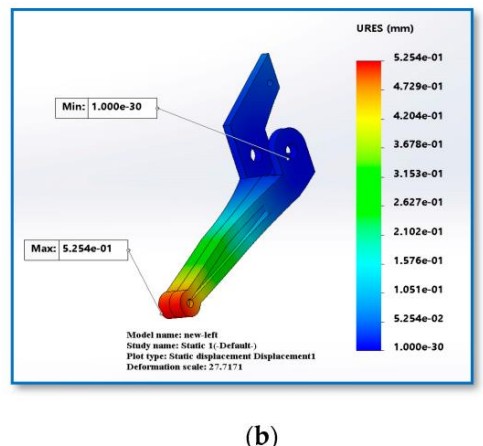

(**a**)        (**b**)

**Figure 13.** (**a**) Displacement analysis diagram for femur; and (**b**) Displacement analysis diagram for tibia.

In the simulation process, we also obtained Factor of Safety diagrams, as shown in Figure 14a,b for the femur and tibia, respectively. The Factor of Safety indicates the safety of any mechanical component during operations; therefore, we can determine how safe the component is under certain conditions, according to the factor of safety scale.

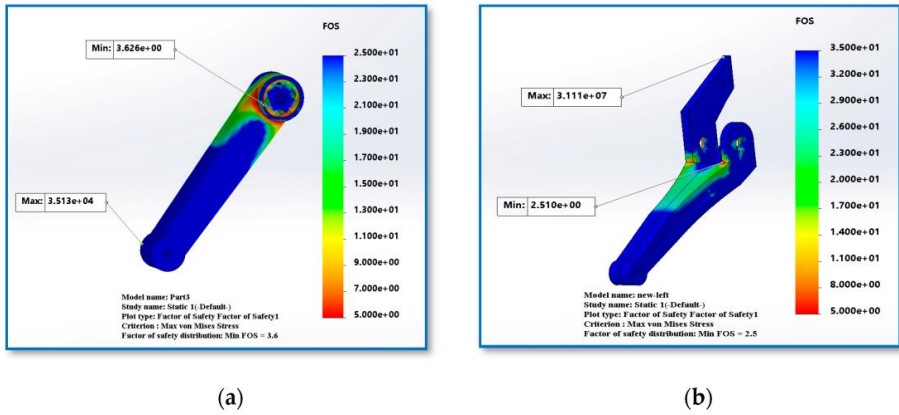

(a)                                                          (b)

**Figure 14.** (**a**) Factor of Safety analysis diagram for femur; and (**b**) Factor of Safety analysis diagram for tibia.

A base number indicates the minimum factor of safety where the component is safe. A value lower than the base number will indicate the failing point of the component. The factor of safety can vary for different materials, and the process of finding the factor of safety varies for ductile and brittle materials. For our ABS Plastic, which is a brittle material, the factor of safety is determined as the ratio of ultimate tensile stress to working stress.

The diagrams show the Factor of Safety scale on the right, which indicates the safety factor of components. Here, the factor of safety for the femur ranged from 3.6 up to $3.5 \times 10^4$, depending on the working stress in different parts of the femur. For the tibia, the factor of safety ranged from 2.510 to $3.111 \times 10^7$. The components are highlighted with multiple colors, indicates the areas where the components are safe (blue area) and where the components are at the risk of failing (red area).

From the simulation process, we also obtained strain analysis diagrams, as shown in Figure 15a,b for the Femur and Tibia, respectively. Strain is associated with the ratio of deformation under load to the original state. The strain is directly proportional to the applied stress. We can analyze the strain condition of our femur and tibia from the load analysis. For the femur, we can see that, with a deformation scale of 1838.92, the strain ranged from $4.004 \times 10^{-9}$ to $4.358 \times 10^{-5}$. For the tibia, with a deformation scale of 27.71, the strain ranged from $3.457 \times 10^{-10}$ up to $3.257 \times 10^{-3}$. The components are highlighted with multiple colors, indicates the areas where strain is more significant. Note that the deformation scale was adjusted for clear visual analysis.

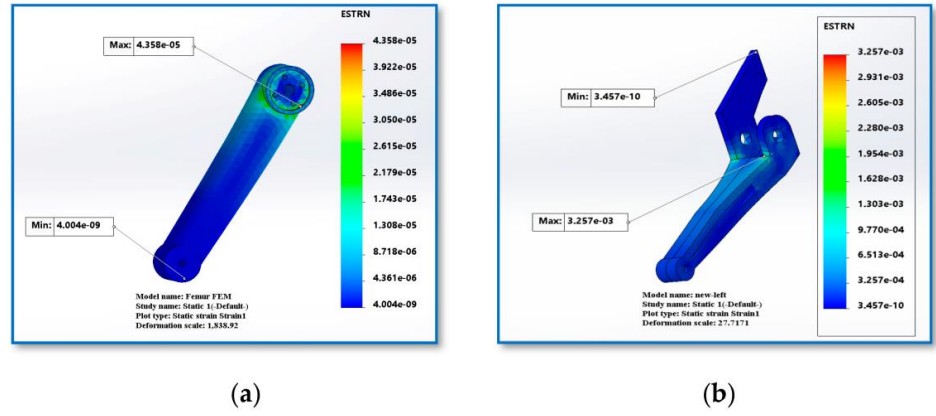

(a)                                                          (b)

**Figure 15.** (**a**) Strain analysis diagram for femur; and (**b**) strain analysis diagram for tibia.

### 4.2. Unified Robot Description Format Verification

The URDF refers to a robot as a tree of links connected by joints. The joints define how one link moves concerning another, thereby defining the location of the links in space. The links represent the moving components of the robot. URDF contains every link's inertia, mass, and rotation axis information. To create a scientific simulation of kinematics after finishing the assembly of the robot, we created a URDF file for the robot, which contains the joint parameter information such as translation and rotation along the x, y, and z axes, as well as roll, pitch, and yaw movements. Table 5 shows the urdf verification parameters.

**Table 5.** Verification at various angles with URDF.

| Robot Positions | Hips Joint Angle (Degree) | | Femur Joint Angle (Degree) | | Tibia Joint Angle (Degree) | |
|---|---|---|---|---|---|---|
| Ideal State Figure 16a | Coxa FR | 0 | Femur FR | 0 | Tibia FR | 0 |
| | Coxa FL | 0 | Femur FL | 0 | Tibia FL | 0 |
| | Coxa BR | 0 | Femur BR | 0 | Tibia BR | 0 |
| | Coxa BL | 0 | Femur BL | 0 | Tibia BL | 0 |
| Standing State Figure 16b | Coxa FR | 0 | Femur FR | −45 | Tibia FR | 90 |
| | Coxa FL | 0 | Femur FL | 45 | Tibia FL | 90 |
| | Coxa BR | 0 | Femur BR | −45 | Tibia BR | 90 |
| | Coxa BL | 0 | Femur BL | 45 | Tibia BL | 90 |

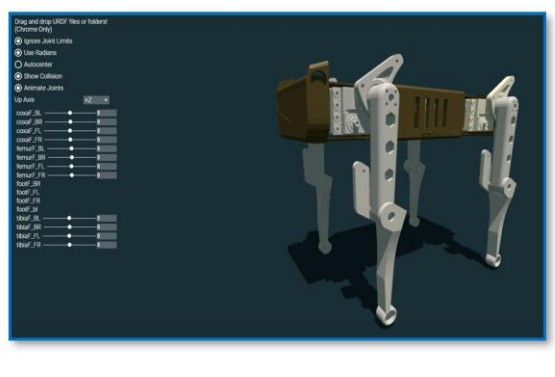 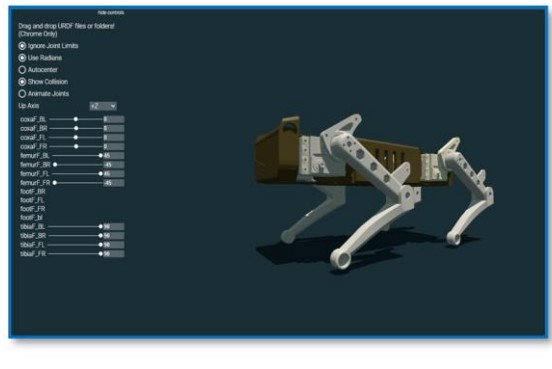

(**a**)      (**b**)

**Figure 16.** (**a**) Ideal state position; and (**b**) standing state position.

In the URDF file, we can utilize visualization and material information to create the simulation effectively. If the URDF file has a problem, the simulation results will be poor. For this issue, after creating the URDF file with meshes, we verified the design using a Web-based URDF visualization tool. Scenario 1 shows the ideal state of our quadruped robot. We created the URDF file in SolidWorks, where we set the positions of all joint angles and links, as shown in Figure 16a. All of the links were set to zero position, relative to the base link (black body). For each leg, the coxa was the parent link, the femur was the child link relative to the coxa, and the tibia was the child link relative to the Femur. Figure 16b shows the standing state of our quadruped robot, where all the links and joint angles were manipulated within their respective parameter ranges; in particular, the coxa remains at zero position, while the femur link is set to 45° relative to the coxa and the tibia link is set to 90° relative to the Femur. The main idea of testing the URDF is to verify the angles of the coxa associated with the other links (i.e., the femur and tibia). Figure 17 shows the URDF tree.

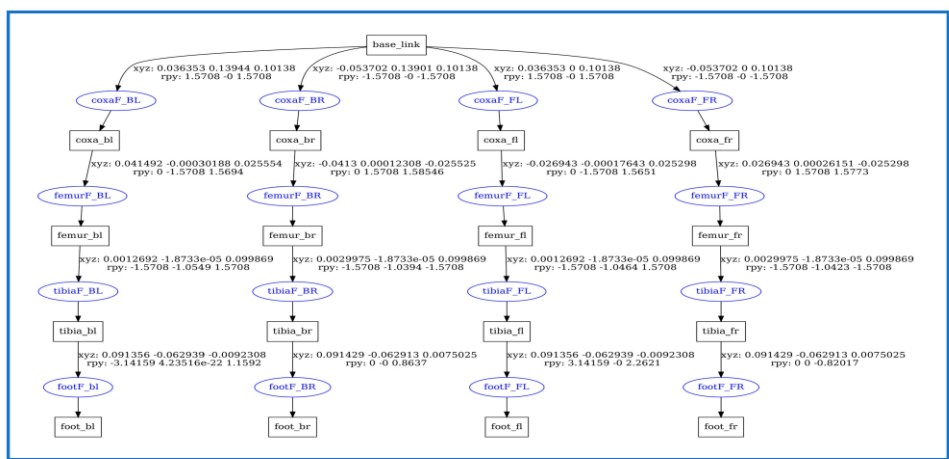

**Figure 17.** Unified robot description format tree. This diagram shows every joint detail with x, y, z and r, p, y values. Each leg has four joints and four links; three joints are revolute, and one is fixed.

*4.3. Dynamic Simulation Results*

We created a quadruped robot using a novel mobility strategy for the leg called the Watt six-bar linkage mechanism. The Pybullet physics engine platform was used for testing of the developed robot, and the robot's dynamic model was established through simulation. The robot was divided into multiple parts, according to its components, and each piece was handled as a separate entity. The maximum torque and speed of each joint, as well as the mass, inertia, and collision geometry of each component, were precisely entered into the simulation environment. In Figure 18, we show our quadruped robot from various angles. We also provide detailed explanations of the mechanism and the kinematic equations with pertinent figures below.

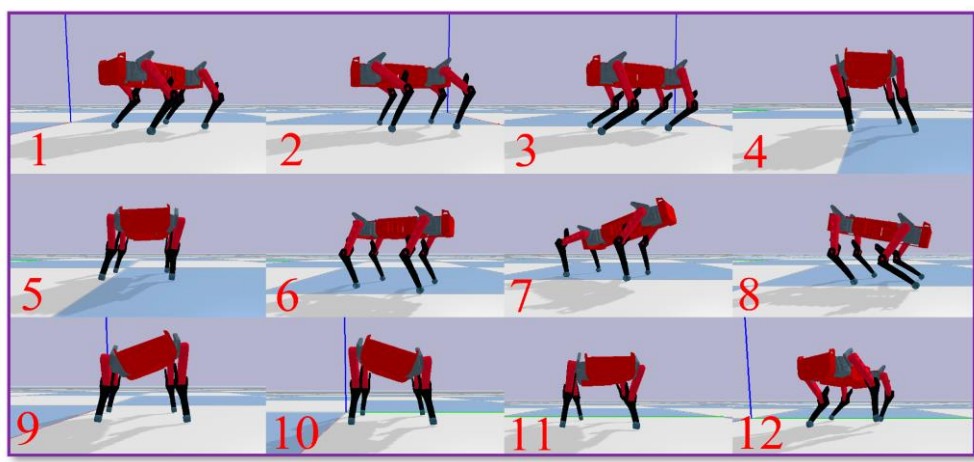

**Figure 18.** Computational simulation results.

Before the simulation process, we connected gait patterns (e.g., trot and creep gait) in our simulation, in order to observe the walking motion of the quadruped robot. In Figure 18, Scenario 1 shows the natural standing position of the quadruped robot. Scenarios 2–6 demonstrate translation of the body of our quadruped robot along its x, y, and z axes. We can see the pitch of the body along its y-axis in Scenarios 7 and 8, while Scenarios 9 and 10 show the roll of the body along its x-axis. Finally, we can see the yaw movement of our robot in Scenarios 11 and 12. Throughout the simulation, we captured several alternative orientations of the robot. Specifically, we demonstrated our quadruped robot's ideal state, roll, pitch, and yaw positions. Following that, we built a real robot that was coupled with a simulation engine. We put our quadruped robot through specific tests, in order to evaluate its pitch position.

Having matched the hardware configuration to the software simulation, we could adjust the pitch position of the actual robot to influence the model in the simulation using the Pybullet Physics Engine. Figure 19a displays the ideal functioning of our simulation model. The sensor's roll and pitch angle data were successfully acquired during our hardware orientation test. We constructed a roll–pitch vs. time graph (Figure 19b), based on the information gathered by our sensors. Our simulation model's pitch orientation is depicted in Figure 19c. We set the robot's pitch orientation on the hardware test-bench. The pitch position roll–pitch vs. time graph is depicted in Figure 19d.

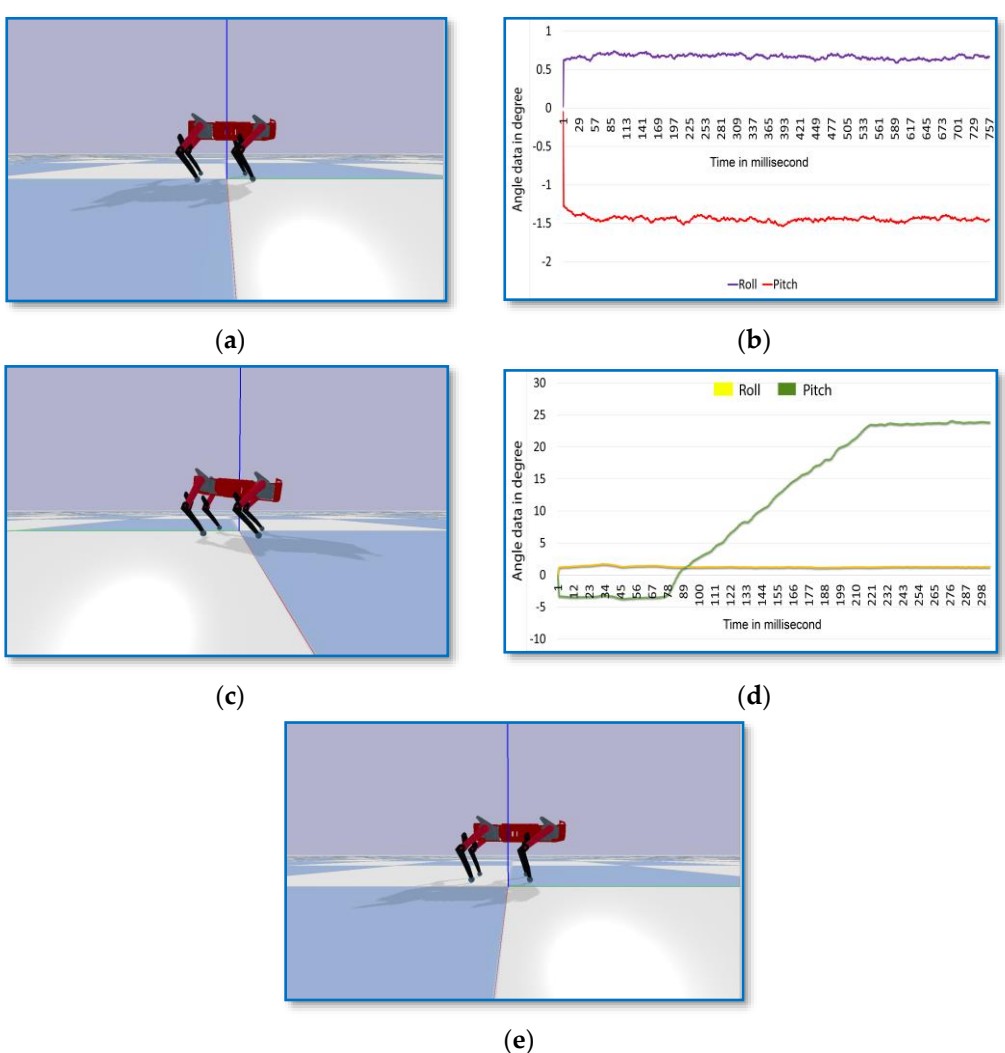

**Figure 19.** (**a**) Ideal state position of robot in side view; (**b**) Ideal state graph; (**c**) Pitch 9 of robot; (**d**) Roll–Pitch vs. time graph; and (**e**) Self-Balanced position.

With hardware and software synchronization, we not only could test how the hardware setup manipulates the simulation model, but also vice versa. We used Proportional Integral Derivative with our kinematics equations. While configuring our robot into a pitch position, the simulated model moved such that its center of mass took precedence over the initial pitch. Figure 19e demonstrates how the simulated model adjusted its femur and tibia to counteract the pitch position. The central body always lies flat against the reference plane in the virtual world. Viewed from the front, our simulation model in its ideal state is depicted in Figure 20a. The sensor's roll and pitch angle data were successfully acquired during our hardware orientation test. Figure 20b shows a graph depicting the roll–pitch angle over time, as measured by our sensors. The roll orientation of our simulation model is displayed in Figure 20c. The related roll–pitch vs. time graph for roll position is shown in Figure 20d.

Figure 20e shows how the simulation model stabilized when we positioned the gyroscope sensor to induce roll. The core coordinate system was maintained in its initial location, while the rest of the model counteracted the roll by shifting the coxa angle. In this way, the simulated model can maintain a level stance relative to the reference plane.

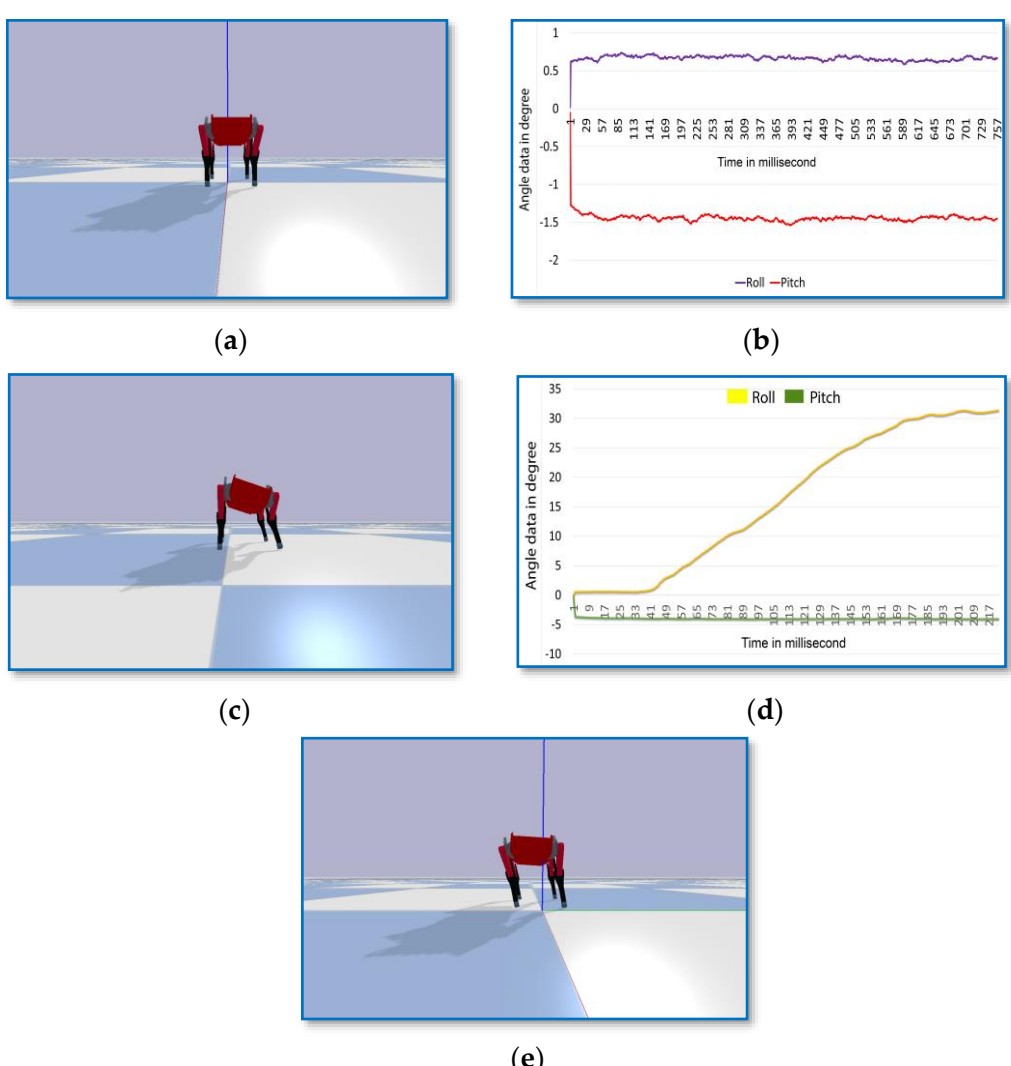

**Figure 20.** (**a**) Ideal state position of the robot in front view; (**b**) Ideal state graph; (**c**) Roll orientations of the robot; (**d**) Roll–Pitch vs. time graph; and (**e**) Self-Balanced position.

We constructed a real robot using 3D-printed components and the six-bar connection system. When we assessed the robot's trot gait pattern, its walking was essentially flawless. Figure 21 clearly depicts the motion of the trot gait pattern. According to the flat terrain test, we simultaneously recorded the roll vs. pitch sensor data as a graph. During the trotting period, the robot recorded a minimum roll angle of $-5°$ and a maximum roll angle of $16°$. The minimum pitch angle was $-11°$, while the highest pitch angle was $9°$. The angle data are displayed in Figure 22. The gyroscope results of the standing robot were $-1.2°$ pitch and $0.5°$ roll. Some photographs of our robot are shown in Figure 23.

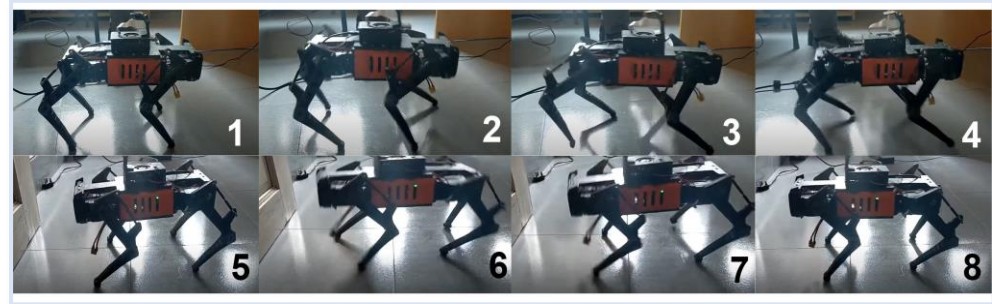

**Figure 21.** Still pictures of adopted watt six-bar linkage mechanism-based quadruped robot. Sequence 1–8 shows the trot gait transition on flat terrain.

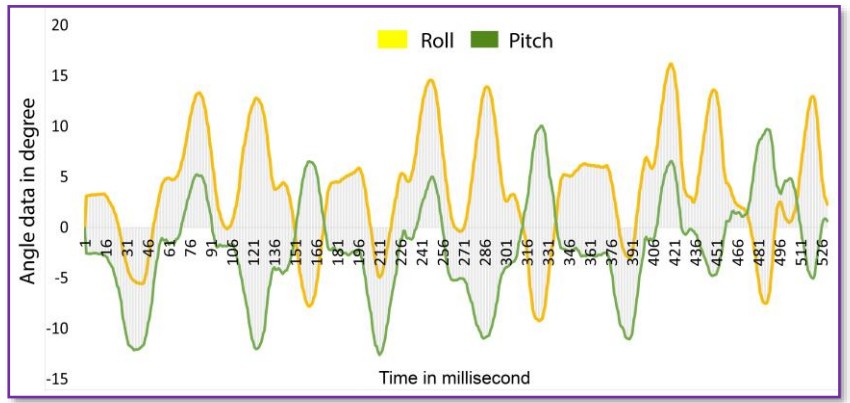

**Figure 22.** Roll pitch data extracted from running trot gait, according to Figure 21 (real robot flat terrain test).

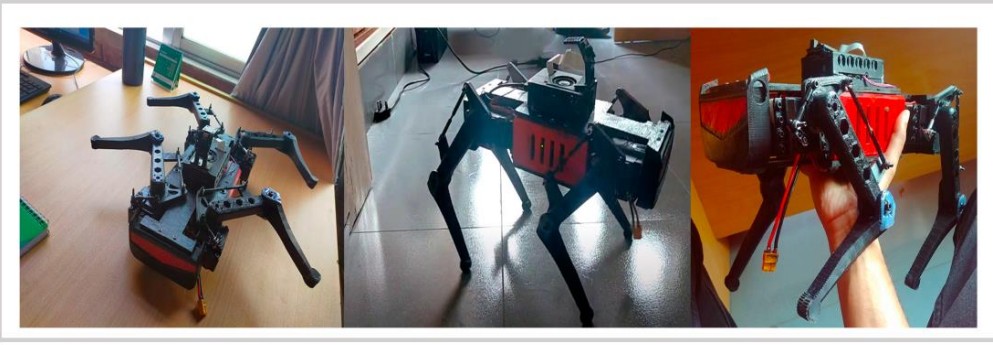

**Figure 23.** Still pictures of the implemented version of Iron Dog mini.

Using the Pybullet physics engine simulation, we achieved dynamic simulation results. Several translational and rotational movements were demonstrated. Kinematics (in particular, Forward and Inverse) aided us in achieving this motion. Forward kinematics provide the positions of the end effectors, while inverse kinematics provide the locations of the joints. Therefore, x, y, and z translation, as well as roll, pitch, and yaw movement can be easily achieved using these two mathematical approaches. We have, thus, briefly described the kinematics portion of the mathematical analysis. During the simulation time, robot walking was achieved through the use of a trot gait pattern and a 14-point Bezier curve. According to the mathematical analysis, gait pattern, and trajectory generation, the outcome of the simulation was satisfactory. Details of the practical implementation and a simulation video can be found elsewhere [35].

## 5. Conclusions and Future Work

This study on the low-cost development and simulation of a quadruped robot, considering a six-bar linkage mechanism, is expected to play an important role in the vigorous development of quadruped robots in the future. Within the scope of this investigation, we presented a process for computational simulation and its underlying mathematical rationale. We developed an innovative concept, providing thorough explanations of the steps followed to generate the simulation model. A cutting-edge Watt six-bar linkage system was added to redistribute the weight better. Overall, we have developed a relatively affordable quadruped robot, allowing researchers to more easily pursue careers in this field. Our four-legged robot costs only USD 350, which is significantly less than other quadruped robots such as the Spot (USD 74,500), Unitree a1 (USD 15,000), and Xiaomi Cyberdog (USD 1599). Compared with existing robots, the proposed robot is lightweight, can go through rough terrain, and can carry loads. We also used relatively cheap servo motors and 3D-printed components.

In comparison to more traditional methods, including the belt-pulley approach utilized in previous quadruped robots, we opted to use the Watt six-bar linkage mechanism, due to its intuitive design. We provided in-depth kinematic analysis data. We may consider training the robot's gait using this model as a template. With any luck, our upcoming study will demonstrate the real-time software and hardware synchronization system, including a reinforcement learning technique. One of the goals of this research is to use AI in a simulation environment, then take the results and use them to fine-tune a physical robot. In Table 6 we exhibited the cost breakdown table.

**Table 6.** Cost Breakdown table.

| Component | Per Unit Cost (USD) | Quantity | Total Cost (USD) |
|---|---|---|---|
| SPT5535LV Actuator-Servo | 15 | 12 Pieces | 180 |
| Metal Servo Arm | 0.50 | 4 Pieces | 2 |
| Metal Servo Horn | 0.40 | 4 Pieces | 1.6 |
| Hex spacer | 0.10 | 30 Pieces | 3 |
| M3 Screws | 0.05 | 85 Pieces | 4.3 |
| Brass Inserts | 0.02 | 85 pieces | 1.7 |
| 16Awg Silicon wire | 1.5 (per yard) | 5 yards | 7.5 |
| 3D printing Abs material | 21.99 (per kg) | 2 kg | 43.98 |
| Raspberry Pi 4b | 40 | 1 Pieces | 40 |
| Bluetooth Joystick Controller | 16 | 1 Pieces | 16 |
| Arduino Pro Mega | 8 | 1 Pieces | 8 |
| 41 A Buck Converter | 5 | 1 Pieces | 5 |
| 7.4 V 2200 mAh Battery | 18 | 1 Pieces | 12 |
| IMU 6050 | 0.6 | 1 Pieces | 0.6 |
| **Total Cost** | | | 332.18$ |

Throughout this research, we designed and built our quadruped robot and developed a kinematic method for its simulation in the Pybullet Physics Engine. We used 3D printing technology to test the model in the actual world when the simulation was complete. With the help of our simulated model, we could correct most issues, including improper scaling and changes that would have harmed the robot. The use of a simulation model makes it straightforward to examine the kinematic model and create a gait. Using the simulation, we can refine our robot model to keep up with the real robot. Our long-term goal is to

advance robotics by creating a low-cost, highly developed quadruped robot that the next generation of scientists and engineers can easily use.

**Author Contributions:** Methodology, M.H.R.; Formal analysis, S.B.A., T.D.M., M.F.U. and M.H.; Investigation, M.H.R. and T.D.M.; Writing—original draft, M.H.R.; Visualization, M.H.R.; Supervision, S.B.A.; Project administration, M.H.R.; Funding acquisition, S.B.A. and M.H. All authors have read and agreed to the published version of the manuscript.

**Funding:** This research received no external funding.

**Institutional Review Board Statement:** Not applicable.

**Informed Consent Statement:** Not applicable.

**Data Availability Statement:** Data is available upon request from the authors.

**Acknowledgments:** We would like to express our appreciation and gratitude to Fayed Al Monir's help with the mathematics and (FEM), (FoS) testing of our research work.

**Conflicts of Interest:** The authors declare no conflict of interest.

## Abbreviations

The Following abbreviations are used in this research article.

| | |
|---|---|
| URDF | Unified Robot Description Format |
| DoF | Degrees Of Freedom |
| FEM | Finite Element Method |
| FoS | Factor of Safety |
| MIT | Massachusetts Institute of Technology |
| CAD | Computer Aided Design |
| STL | Stereolithography |
| PID | Proportional Integral Derivative |
| IMU | Inertial Measurement Unit |
| FR | Front Right |
| FL | Front Left |
| BR | Back Right |
| BL | Back Left |
| g | gram |
| Cm | Centimeter |

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
