# Peer review of "A Dynamic Approach to Low-Cost Design, Development, and Computational Simulation of a 12DoF Quadruped Robot"

_robotics, doi:10.3390/robotics12010028_

Round 1

Reviewer 1 Report

The subject of the paper is relevant and the design procedure is sound but the focus of the paper is not clear: the advantage of the Watt linkage for leg actuation in comparison with the 4 bar linkage is not shown and the cheapness of the realization is claimed but not demonstrated. In other words, the work is correct but scarcely original and lacks an experimental validation of the claimed features.

Minor remarks:

* English language should be revised.

* Equations should be numbered

* The Watt linkage is not novel as claimed in the abstract

* Too many references are made from regional Conferences or journals with scarce visibility

* Introduction and review of the state-of-the-art should be more focused on the subject of the paper

* The Aibo project by Sony dates back to 1999 (see Tab. 1)

* The abbreviation of “grams” is “g” not “gm”

* Most figures are not referenced in the text and they are often scarcely explanatory. 

* The names of the links in table 2 are not clearly defined and explained within the article. 

* Figure 4a is obscure: mechanisms kinematics, motion driver and frame hinges are not easily identified. 

* In Figure 5 the distance L1 and the angle θ1 are not represented and the frame 1 is not clearly positioned 

* After equation 1 the triangle is not ADE but ADC, same thing after equation 7 where the letter F is not present.  

* In the equation yielding θ2 it is not clear whether one should take as reference the plane where the leg lies (given by the value of θ1) or a plane parallel to the plane X_0 Y_0

* The formula providing the value of θ3 is wrong: +L12 should be substituted by -L12 

* In (6) the power and the subscript of x and y is wrong

* The use of boldface in some expressions of θ is not correct 

* The simulation model workflow part requires more explanation and illustration in detail.

* The FEM analysis is scarcely reported: the colour map values are not readable, the application of the 100 N load is scarcely explained, the physical meaning of the safety factor is obscure nor it is explained how it is calculated. Same considerations for the strain. 

* The results of the dynamic symulations are not sufficiently commented and appropriate conclusions are not drawn.  

Reviewer 2 Report

The paper presents the design, prototyping and simulation of a 12dof quadruped robot - Iron Dog mini. 

- Throughout the article, the acronyms are never defined (URDF/DOF/MIT/...); 

- You say that "The purpose of this research was to develop a low-cost, modular, quadruped assistive-robot platform for use in security and surveillance operations" is a contribution of your work, but this is the motivation, not the contribution. Please review;

- Figures are sometimes pixelized or the size is unformatted, please review;

- Why are there some equations bolded and others don't? It doesn't make any sense. Please follow the mathematical protocol for equations/matrices. Also, all the equations (even the ones derived from the previous ones) must be numbered and mentioned on the text with an explanation;

- On equation 5, you have "? ????s", rather than arccos;

- Some Tables/Figures appear with the 0 before the number (e.g. Figure 02), other don't (e.g. Figure 3). This must be reviewed;

- On Table 2, the mass is mention on gm, while it should be on g (grams). Also, it is easier and more understandable for the reader if the measurement units appear on the title (e.g. mass (g));

- There must exist a table with the complete dimensions/weight of the robot, fully mounted;

- The article does not present a clear photo of the physical robot, rather simulated ones. I think this is important;

- Figure 3 is confusing. The arrows should not cross the robot, rather be putted in its right side;

- If the robot is low-cost and can be helpful to the scientific community, I think that the printable parts must be available on an open-source repository;

- Some references are mention with the number, like "[13] has highlighted 2dof-based parallel leg formation". This can't be done, here it must name the authors, for example;

- The numbers of the references, figures, tables, etc. aren't linking to the proper label on the pdf;

- In the command block of Figure 9, it has an error on Pitch;

- Table 6 must provide the links to the places where the hardware can be bought, considering the scientific community can replicate your work;

- Also considering Table 2 your robot weighs 1190 g with actuators, but you have used 2kg of ABS. How is this possible? 

- Regarding the control, are you just using PIDs on every joint? The block diagram for the control mechanism should be included and well explained, including the PID constants.  

- A movie must be provided, considering the work with the physical robot;

Finally, I think that the article is suitable for the journal.
However, several typos must be addressed, so authors must re-read the manuscript.
In my opinion, it has a high potential, but the manuscript should undergo major revisions for it to be able to be accepted. 

Round 2

Reviewer 1 Report

The manuscript is now much clearer and its quality has been improved. However my original concerns have not been overcome, e.g. about the novelty of the concept, the advantages of a more complex actuation (a six bar mechanism in place of a 4 bar linkage, especially when claiming a cheaper realization) and the need for an experimental proof of the claimed performance.

Some minor flaws are still present, e.g. the unnecessary long mathematical passages (see eq. 2-12) or the explanations of trivial concepts, the need of a complex 14 points Bezier curve to plan the gait, the full transform matrix (1) in place of the usual simpler form, etc.

Reviewer 2 Report

I want to thank the authors for considering all my suggestions and changing the document accordingly. 
